# VLP-ELISA for the Detection of IgG Antibodies against Spike, Envelope, and Membrane Antigens of SARS-CoV-2 in Indian Population

**DOI:** 10.3390/vaccines11040743

**Published:** 2023-03-27

**Authors:** Dilip Kumar, Sourav Singha Roy, Ruchir Rastogi, Kajal Arora, Avinash Undale, Reeshu Gupta, Nupur Mehrotra Arora, Prabuddha K. Kundu

**Affiliations:** 1Research and Developmental Laboratory, Premas Biotech Private Limited, Sector 4, IMT Manesar, Gurgaon 122050, Indiagreshu12@gmail.com (R.G.);; 2Centre of Research for Development, Parul University, Vadodara 391760, India

**Keywords:** COVID-19, immunoglobulin G, virus-like particles, ELISA, nanoparticle tracking analysis

## Abstract

Background: Serological methods to conduct epidemiological survey are often directed only against the spike protein. To overcome this limitation, we have designed PRAK-03202, a virus-like particle (VLP), by inserting three antigens (Spike, envelope and membrane) of SARS-CoV-2 into a highly characterized *S. cerevisiae*-based D-Crypt™ platform. Methods: Dot blot analysis was performed to confirm the presence of S, E, and M proteins in PRAK-03202. The number of particles in PRAK-03202 was measured using nanoparticle tracking analysis (NTA). The sensitivity of VLP-ELISA was evaluated in 100 COVID positive. PRAK-03202 was produced at a 5 L scale using fed-batch fermentation. Results: Dot blot confirmed the presence of S, E, and M proteins in PRAK-03202. The number of particles in PRAK-03202 was 1.21 × 10^9^ mL^−1^. In samples collected >14 days after symptom onset, the sensitivity, specificity, and accuracy of VLP-ELISA were 96%. We did not observe any significant differences in sensitivity, specificity, and accuracy when post-COVID-19 samples were used as negative controls compared to pre-COVID-samples. At a scale of 5 L, the total yield of PRAK-03202 was 100–120 mg/L. Conclusion: In conclusion, we have successfully developed an in-house VLP-ELISA to detect IgG antibodies against three antigens of SARS-CoV-2 as a simple and affordable alternative test.

## 1. Introduction

The asymptomatic transmission of SARS-CoV-2 has raised the need for an accurate epidemiological survey of SARS-CoV-2 to manage the different phases of the COVID-19 pandemic. The initial estimates of asymptomatic infection rates from the West were approximately 40–45%. While rRT-PCR is the gold standard and unanimous choice, in many countries, it is not used for epidemiological surveys because of its high cost, especially in developing countries. Epidemiological surveys will be useful in making several public health decisions related to the coronavirus disease (COVID-19) vaccination campaign and the World Health Organization (WHO) COVID-19 Vaccines Global Access (COVAX) Program [1,2]. Due to cost and time factors, serological assays that are positively associated with true antibody concentration (IgM/IgG) can be an important approach to identify asymptomatic patients by conducting epidemiological surveys. Additionally, these assays can also be used for COVID patients with negative rRT-PCR results in the later stages of infection [3,4]. Serological assays, such as lateral flow assays (LFAs) and enzyme-linked immunosorbent assays (ELISAs), are valuable tools for epidemiological surveys, helping to understand the spread of the disease, the impact of containment measures, monitoring herd immunity, predicting risk, and prioritizing groups for vaccination [5,6,7]. A wide range of serological immunoassays that complement rRT-PCR uses different SARS-CoV-2 antigen targets and formats [8,9,10]. The most common antigens are spike glycoprotein S1 with a receptor-binding domain (RBD), nucleocapsid protein, or both [11,12]. Membrane proteins also play an important role in recognizing IgM and IgA in SARS-CoV-2-infected patients. High levels of antibody responses have been observed in patients with COVID-19, with moderate and severe forms of the disease [13]. The performance of these assays varies owing to the choice of antigen, nature and structure of the antigen, or disparity in patients [14,15,16]. However, serological tests targeting one antigen were found to generate false-negative data in approximately 40% of the populations tested [17]. Depending on the isotype tested, false-negative data can further increase. False-negative results are more harmful to society because infected individuals cannot be isolated and can infect others. Additionally, most of the serological assays detect antibodies against spike or nucleocapsid proteins and have low (27–41%) sensitivity in the first week, which increases to 78–88% in week 3 [18] However, it has been shown that the IgG-specific antibody response against membrane proteins of SARS-CoV-2 does not change over time [19]. Similarly, antibodies against envelope proteins have been identified in COVID patients during active infection [20]. These results suggest that antibodies against E and M antigens develop during the early stages of infection. This indicates that the sensitivity and specificity of serological assays can be enhanced by identifying IgG antibodies against E and M antigens during the early stages of infection [12,21]. In addition, the simultaneous detection of several antigens can predict disease outcomes by identifying various antibody signatures, allowing for the assessment of the immunological response against future vaccines with greater objectivity.

Here, we developed a VLP-ELISA to detect immunoglobulin G (IgG) antibodies against highly pathogenic SARS-CoV-2 using PRAK-03202 (VLP) in patient serum samples. The design and manufacture of the PRAK-03202 represent a plug-and-play process in which we insert the three-target antigens (S, E, and M) sequence into a highly characterized *S. cerevisiae*-based D-Crypt™ platform [22]. D-Crypt is a protein expression platform designed for high-yield production of ‘difficult-to-express’ proteins (DTE-Ps). The D-Crypt platform combines a yeast expression host with more than 20 custom-made expression vectors. Introducing additional structural proteins, such as E and M proteins, as antigens in VLP-based ELISA will lead to efficient epidemiological surveys against variants of SARS-CoV-2. The inclusion of E and M proteins will also enhance the sensitivity of the assay during the early stages of SARS-CoV-2 infection. The current study is a follow-up to our previous study in which we demonstrated the therapeutic role of PRAK-03202 against SARS-CoV-2 infection [22]. This study aimed to demonstrate the significance of PRAK-03202 in developing a VLP-ELISA to detect IgG antibodies directed against S, E, and M proteins of SARS-CoV-2 in rRT-PCR positive serum samples. 

## 2. Materials and Methods

### 2.1. Dot Blot Assay

Dot blot analysis was performed to confirm the presence of the S, E, and M proteins in PRAK-03202. A polyvinylidene fluoride (PVDF) membrane was prepared by soaking the membrane in methanol for 20 s. Next, the membrane was equilibrated in the transfer buffer for 5 min. For dot blotting, ~20 μg of the sample was spotted onto a PVDF membrane at the center of the grid. The membrane was blocked with blocking buffer (5% skim milk in 1X tris-buffered saline, 0.1% Tween 20 (TBST)) for 1 h at room temperature. Three washes with TBST were performed and the membrane was probed using 1:1000 dilution of S-specific antibody (Cat#ZHU1076, Sigma-Aldrich, St. Louis, MO, USA) and 1:500 dilution of polyclonal M- (Cat# GTX134866, GeneTex, Irvine, CA, USA) and E- (Cat#MBS150849, MyBioSource, San Diego, CA, USA) specific antibodies. After three more washes with TBST, the membrane was incubated with an HRP-conjugated secondary antibody (1:4000) for 1 h at 37 °C. Signals were detected with ECL solution (Bio-Rad cat#170-5061) using the Immuno-Blot Development Instrument (G: Box, SYNGENE, Cambridge, UK).

### 2.2. NTA Analysis

PRAK-03202 was diluted at two different concentrations (0.25 and 0.5 mg) in potassium phosphate buffer (pH = 7.5) and outsourced to IIT Hyderabad for nanoparticle tracking analysis (NTA). NTA was performed using a NanoSight LM10 instrument (NanoSight, Amesbury, UK) to measure the number of particles in PRAK-03202 preparations. 

### 2.3. Quantification of S, E, and M by ELISA

ELISA was performed to quantify S, E, and M proteins in PRAK-03202. In-house-developed purified proteins (S, E, and M; Source: Premas Biotech, IMT Manesar, Gurugram, India) were coated overnight on ELISA plates in the range of 1000 ng to 7 ng per well. The purity of the in-house-developed S, E, and M proteins was tested using SDS-PAGE (Appendix A). Next, 1% bovine serum albumin (BSA) solution was used for blocking, and the plates were incubated with primary antibodies for S, E, and M proteins at dilutions of 1:250, 1:250, and 1:500, respectively. After washing, HRP-conjugated goat anti-rabbit secondary antibody (PI31460, Fisher Scientific, Waltham, MA, USA; Dilution-1:4000) was applied. Finally, the chromogenic reaction was quantified following the addition of 3,3′,5,5′-Tetramethylbenzidine (TMB) substrate (Invitrogen) and stop solution (1N H_2_SO_4_). The absorbance of the samples was measured at 450 nm and a standard curve was constructed. A similar procedure was repeated for PRAK-03202, where plates were coated with 250 ng of PRAK-03202 for S and M proteins and 1000 ng of PRAK-03202 for E protein. After blocking with 1% bovine serum albumin (BSA), the plates were incubated with S, E, and M antibodies. After washing, the respective secondary antibodies were applied, and the absorbance was measured at 450 nm after adding TMB and stop solution. The concentrations of S, E, and M proteins were calculated from the standard curve. The numbers of S, E, and M molecules in PRAK-03202 were determined using an online calculator (https://www.bioline.com/media/calculator/01_04.html, accessed on April 2021).

### 2.4. Production of PRAK-03202 at 5 Liter Scale

Production of PRAK-03202 was divided into two parts: upstream and downstream processes.

(i.)Upstream processing

An amount of 500 μL glycerol stock of yeast host cells expressing PRAK-03202 was inoculated directly onto 500 mL of selective seed media (6.7 g/L yeast nitrogen base without amino acids, 1.40 g/L, yeast synthetic drop-out media without uracil and leucine, 0.04 g/L tryptophan, 0.02 g/L histidine, 20 g/L D-glucose). Three batches (7, 8, and 9) were used to observe consistency in the process. The diluted seed media was incubated for 20–24 h at 250 rpm and 28 °C until the OD of the media reached approximately 80. The well-grown seed medium was aseptically transferred to a fermenter containing a sterilized fermentation medium (20 g/L Hi veg peptone and 10 g/L yeast extract). The critical process parameters (CPP) for fermentation were maintained at 28 °C ± 1, 5.8 ± 0.2 pH, 30 ± 10% dissolved oxygen (DO), 150–950 revolution per minute (RPM) agitator speed, 0.5–1.0 vessel volumes per minute (VVM) of airflow, and 60–62 h of fed-batch fermentation time. The CPP for the induction phase was set to a wet cell weight of 50 ± 5 g/L. The agitator speed and airflow were varied to maintain a DO of 30 ± 10%. The fermenter was harvested at 24, 48, and 72 h to determine the optimum time to obtain the maximum concentration of PRAK-03202. After centrifugation, the pellet and supernatant of the cultures were stored separately at −70 °C until further downstream processing.

(ii.)Downstream processing

Briefly, 5 L of cell pellets from the fermentation batch were dissolved in lysis buffer containing benzonase nuclease (100 mM potassium phosphate pH 7.2, 0.001% Tween 80, 2 mM PMSF) at a ratio of 1:3:3, followed by homogenization at 600 bars and five passes at 4 °C. The homogenate was centrifuged for 15 min at 8000 relative centrifugal force (RCF), incubated for 18 h at 4 °C, and finally centrifuged at 16,000 RCF for 45 min. The supernatant was clarified by micro-, ultra- (750 kDa hollow fiber), and diafiltration (750 kDa hollow fiber) and followed by column chromatography (size exclusion chromatography–HPLC) to detect the presence of PRAK-03202. The diafiltered retentate was loaded onto a Capto Core 700 column pre-equilibrated with diafiltration buffer (20 mM potassium phosphate pH 7.2, 500 mM KCl). Fractions were pooled based on VLP purity, dialyzed with formulation buffer (100 mM potassium phosphate pH 7.2, 0.001% Tween 80, 5% sucrose), and sterilized using a 0.2 µm filter. The sterilized fractions were tested for final quality parameters, such as sterility testing and analytical studies.

### 2.5. Multiple Sequence Alignment of BQ.1.1 and Wuhan Variant of SARS-CoV-2

Multiple sequence alignment of BQ.1.1 (Accession: OQ291482.1) and Wuhan Variant (Accession: NC_045512) for SARS-CoV-2 was performed using Clustal Omega [23].

### 2.6. Human Serum

We used a panel of 155 samples to validate our assay. Out of 123 samples, 23 serum samples were collected before the COVID-19 pandemic, and 32 rRT-PCR negative samples were collected during the post-COVID era. All non-SARS-CoV-2 sera samples before the COVID-19 era were gifted by the AIIMS (Delhi), RML Hospital (Delhi), and the National Institute of Tuberculosis and Respiratory Disease (Delhi). The 100 rRT-PCR positive and 32 post-COVID rRT-PCR negative samples were provided by the biorepository of the Translational Health Science and Technology Institute (TSHTI), Faridabad. The information on the number of days since symptom onset was retrieved from medical records. The patient cohort of the biorepository received institutional ethics committee approval (THS 1.8.1/(107) dated 15 January 2021). The study protocol was conducted in accordance with the principles of the Declaration of Helsinki. Samples were stored in the laboratory at −20 °C until analysis. Frozen samples were thawed for one hour at room temperature on the day of analysis. The thawed samples were vortexed before analysis.

### 2.7. ELISA

This study measured PRAK-03202-specific IgG-mediated antibody titers in serum samples using ELISA [22]. For the PRAK-03202-specific IgG response, wells were coated with 0.2 μg of PRAK-03202 proteins at 2–8 °C overnight. Subsequently, the plates were blocked with 1% BSA dissolved in 1 × TBST for 1 h at room temperature. Next, the diluted sera (1:1000) from convalescent patients, pre-COVID negative samples, and post-COVID rRT-PCR negative samples were applied to each well and incubated at room temperature for 1 h. The plates were then incubated with horseradish peroxidase (HRP)-conjugated anti-human secondary antibodies (1:10,000 dilution; A0170, Sigma-Aldrich, USA) at room temperature for 1 h. The complex formed by the bound conjugate was visualized by adding substrate tetramethylbenzidine, which locally produces a soluble, blue-colored product. The reaction was stopped using 2N H_2_SO_4_, and the plates were read at 450 nm using an ELISA plate reader (Multiskan SKyHigh, Thermo Fisher Scientific, Waltham, MA, USA). 

### 2.8. Calculation of Cut-Off, Sensitivity, Specificity, and Accuracy

To determine the cut-off for PRAK-03202, we first calculated the mean optical density (OD) of the COVID-19-negative samples and then applied the following formula: Cut-off = Mean + 3SEM. Furthermore, we plotted the receiver operating characteristic (ROC) curves to observe the performance of our assay. Sensitivity was defined as the proportion of correctly identified COVID-19-positive patients who were initially tested positive using rRT-PCR by detecting SARS-CoV-2 in respiratory samples and those with COVID-19 symptoms. Specificity was defined as the proportion of naïve patients classified as negative. The 100 COVID-19-positive specimens used for the sensitivity analysis were divided into four different groups based on the number of days since symptom onset: 0–7 days, 8–14 days, 15–21 days, and over 21 days, with 25 sera in each group. The ROC curve and other statistical analyses were performed using OriginPro (version 2020b).

## 3. Results

### 3.1. PRAK-03202 Production and Characterization

#### Dot Blot Assay and Particle Analysis of PRAK-03202

Co-expression of S, E, and M proteins in PRAK-03202 was confirmed using a dot blot assay (Figure 1A). Previously, we confirmed the presence of S, E, and M proteins in PRAK-03202 using Western blotting [22]. The total number of physical particles in the sample was quantified by nanoparticle tracking analysis (NTA). Two different dilutions (0.25 and 0.5 mg) of PRAK-03202 were used, which showed linearity of particle counts during the determination of the number of particles (Figure 1B). The NTA of our samples indicated that PRAK-03202 preparations contained 1.21 × 10^9^ mL^−1^ particles, which is approximately equal to the heat-inactivated SARS-CoV-2 virus stock, as reported previously [24]. Next, we developed an ELISA to quantify the numbers of S, E, and M molecules in PRAK-03202. The results showed that the number of S, E, and M molecules in PRAK-03202 was 1.4 × 10^11^ ± 0.32, 1.1 × 10^11^ ± 0.3, and 1.51 × 10^12^ ± 0.05 (ratio: 1:1:10), respectively (Table 1). The maximum concentration of PRAK-03202 was obtained after 24 h in the fermenter (Figure 1C). The total yield was 100–120 mg/L at a 5 L scale. Table 2 shows the analytical characterization of the purified PRAK-03202 obtained at a 5 L scale.

Sequence comparison of BQ.1.1 and Wuhan variant of SARS-CoV-2. BQ.1.1 is a subvariant of Omicron (B.1.1.529). In comparison to the Wuhan variant, the deletion of 3 amino acids (glutamate, arginine, and serine) at position 31–33 and a non-conservative mutation (lysine) at position 13 were observed in the nucleocapsid protein of BQ.1.1. We also observed one non-conservative mutation (isoleucine) at position 9 in the envelope protein of BQ.1.1. However, no non-conservative or semi-conservative mutation or deletion/insertion of amino acids was observed in the membrane protein of BQ.1.1, when compared to the Wuhan variant. As expected, there were many non-conservative, semi-conservative, and addition/deletions of amino acids in the spike protein of BQ.1.1, compared to the Wuhan variant (Figure 2). These results suggest the conserved nature of membrane and envelope proteins compared to other structural proteins in variants of SARS-CoV-2, and, therefore, will increase the sensitivity of the assay during the first and second week of symptom onset by detecting IgG against E and M antigens of both variants.

### 3.2. Determination of the Cut-Off Value for VLP-ELISA Assay

Previous reports have indicated that a steady state of IgG response to a viral infection is reached at 2 weeks after exposure [12,25]. We selected a subset of samples collected >14 days (*n* = 50) after the onset of COVID-19 symptoms to determine the clinical performance of our assay. The preliminary cut-off value of this developed ELISA was 0.83. The cut-off value is high and may be due to reactivity in the pre-COVID sera (possibly due to immune responses against other coronaviruses). The mean OD in the COVID-19 and negative groups was 1.86 ± 0.02 and 0.38 ± 0.15, respectively. The results showed that the IgG response of the patient samples was more than two-fold higher than the cut-off values (Figure 3A). ROC analysis showed satisfactory performance (Figure 3B). After ROC analysis, the area under the curve (AUC) was 0.99. The sensitivity, specificity, and accuracy of the assay after two weeks (>14 days) of symptom onset was 96%. The assay had a negative predictive value of 96% and a positive predictive value of 97%.

### 3.3. IgG Response with Respect to Time after Symptoms Onset

Next, we stratified the capacity of our assay to detect IgG antibodies directed against PRAK-03202 by the time after symptom onset. We defined 4 categories: samples collected less than a week (≤7 days, *n* = 25), between 1 and 2 weeks (8 days and ≤14 days, *n* = 25), between 15 and 21 days (>14 and ≤21, *n* = 25), and more than 21 days (*n* = 25) after symptoms. For the specificity calculation, we used 23 pre-pandemic sera as true-negative sera. The results showed that the sensitivity, specificity, and accuracy of PRAK-03202 were 96%, 96%, and 97%, respectively, at >14 days after symptom onset (Table 3). However, the sensitivity of the assay was lower in the first two weeks of symptom onset than in the third and fourth weeks.

### 3.4. Comparison of Specificity in Pre- and Post-COVID Negative Controls

Previous studies have reported that rapid point-of-care tests have low sensitivity when post-rRT-PCR negative samples are used as negative controls [26,27,28]. To validate this, we compared the sensitivity, specificity, and accuracy of our assay with the pre-COVID (*n* = 23) and post-COVID era (*n* = 32) rRT-PCR negative samples (Table 3). Mean OD in the pre-COVID-19 and post-COVID-19 rRT-PCR negative groups were 0.38 ± 0.15 and 0.53 ± 0.16, respectively. The preliminary cut-off value of this developed ELISA for post-COVID-19 era was 1.03. The sensitivity of the assay was lower when post-COVID-19 samples were used as negative controls compared to pre-COVID-19 negative controls (92% vs. 96%); however, the value was not significantly different (Figure 4A,B).

### 3.5. Comparison of IgG Titers in Early and Later Phases of COVID-19 Symptoms

Another major concern in the antibody response to SARS-CoV-2 is the IgG titer. To determine IgG titers to COVID-19 antigens, we randomly selected eleven samples from the early phase of COVID-19 symptoms (≤14 days) and ten others from later stages (>14 days) and tested serial dilutions of these samples until negativation. Results from these titration curves showed that (Table 4) overall and as expected, IgG titers of samples from the later phase were higher than those from the earlier phase. Three weeks or later after symptom onset, 6/10 of the tested samples presented IgG titers above 50,000 against PRAK-03202. This proportion was 2/11 for samples collected two weeks after symptom onset (Table 4). 

## 4. Discussion

Most of the current serological assays for the detection of SARS-CoV-2 antibodies on the market target one antigen, with some targeting up to two antigens [29]. Currently, many anti-SARS-CoV-2 diagnostic assays in the market target N antigens (Roche, Euroimmun, and Abbott) or S1 antigens (Euroimmun), whereas assays developed by DiaSorin target combination of S1 + S2 antigens. One test used a combination of S, N, and M antigens (United Biomedical UBI SARS-CoV-2 ELISA). However, the results of different tests using the same antigens are not correlated [16]. The ability of IgG, IgM, and IgA to bind membrane proteins in SARS-CoV-2-infected patients has been previously described [13]. We chose the PRAK-03202 as an antigen because it contains three structural proteins (spike, envelope, and membrane proteins) of SARS-CoV-2 [22]. The calculated specificity of PRAK-03202 was 96%, which was lower than that obtained using the Roche assay (i.e., N antigen: 100%); however, it was comparable to the Euroimmun (i.e., N antigen: 96.2%) and DiaSorin (i.e., S1/S2 antigen: 97.7%) assays. The utilization of the three antigens of PRAK-03202 in one assay is in line with the CDC’s recommendations regarding the orthogonal testing strategy. 

The timing of antibody tests plays an important role in the diagnosis of COVID-19. Current data on the immune response to SARS-CoV-2 suggest that seroconversion occurs between 7 and 14 days after the onset of symptoms [11]. However, it is difficult to determine whether the antibody response induced by SARS-CoV-2 will persist over time, for how long, and if they will be protective upon re-exposure to the same or a related virus. In a recent meta-analysis, 178 serological studies on the detection of COVID-19 were included from five different continents (Europe, Asia, North America, South America, and Australia) [18]. This study showed that the sensitivity of several antibody tests was low (27–41%) in the first week, which increased to 78–88% in week 3. Tests that specifically detected IgG or IgM antibodies were the most accurate, and when testing people from 21 days after the first symptoms, they detected 93% of people with COVID-19. Our results also showed that a steady state of IgG response to viral infection was reached 14 days after symptom onset. However, the sensitivity of our assay was higher in the first week of symptom onset than that reported in previous studies. The early detection of antibodies may be useful for understanding the heterogeneity of clinical presentations.

The current study also observed that 96% of the patients had IgG antibodies after the second week of symptom onset. The sensitivity of 96% (2 of the false-negative samples out of 50 samples) after 14 days of symptoms onset was comparable to those obtained on Roche (87.5–100%), Euroimmun (88–95.8%), and DiaSorin (83.3–94.6%) assays. This study suggests that a sample with antibodies against the three antigens is unlikely to yield a false-negative result. French and international health authorities recommend that serological diagnostic assays should have a clinical sensitivity of 90% or more [12]. Our assay largely fulfils these criteria.

Previous studies observed variability in performance among commercially available rapid tests due to differences in validation protocols, with some studies using archived pre-COVID emergence samples [30,31,32] and others using PCR negative samples as negative controls [26,27,28]. Contrary to these studies, we did not observe any significant difference in sensitivity when post-COVID sera were used as a negative control compared with pre-COVID archived samples. This could be due to the presence of three antigens in PRAK-03202 and, therefore, a high IgG response by patients. Although we observed high titers at later phases compared to the early phases, it is unclear whether this IgG is neutralizing.

This study demonstrated that multiplexing several antigens to deliver a result can have high benefits, especially for epidemiological surveys, which will help in planning, implementing, and evaluating public health interventions and programs. This means answering common questions of government such as “What is the status of antibodies in the population?” and “What are the prioritizing groups for vaccination?”. At a country level, governments can judge the status of the protection of citizens and point out gaps and challenges using real-world data. We were also able to produce PRAK-03202 on a larger scale, which would be helpful in conducting epidemiological surveys. The multiplex serology tests developed in this study will reduce the cost per test. Such tests can be used not only for disease surveillance but also as a catalyst to open up the country, monitor herd immunity, and help in safe mobility like other serological assays. This will provide insight into the levels of possible protective immunity and the actual mortality rates, including the proportion of asymptomatic and mild cases. Additionally, due to the limited surveillance data, escaped immunity of the virus by vaccines, and risk of developing infections by variants, this multiplex in vitro assay is anticipated to witness significant growth due to the presence of envelope and membrane protein antigens of PRAK-03202. It should be noted that, whereas mutations in spike proteins can give rise to various variants of SARS-CoV-2, mutations in envelope and membrane proteins are not common (Figure 2).

## 5. Conclusions

The VLP-ELISA developed in this study will reduce the cost per test and will be useful for conducting sero-epidemiological studies in India and elsewhere. The inclusion of E and M proteins will also enhance the sensitivity of the assay during the early stages of SARS-CoV-2 infection. However, the VLP-ELISA assay has several limitations such as (1) the test cannot be intended for COVID diagnosis because IgG is present in the sera of people who have recovered from the disease, and (2) it cannot discriminate between recovered patients and vaccinated patients. (3) IgM response was not evaluated. (4) Unspecific reactions by other coronaviruses and other pre-existing infections or diseases such as malaria, autoimmune, and rheumatic diseases causing false-positive results in ELISAs have not been evaluated. All these evaluations must be performed before this assay and should be used for any studies analyzing patients’ immune response.

## Figures and Tables

**Figure 1 vaccines-11-00743-f001:**
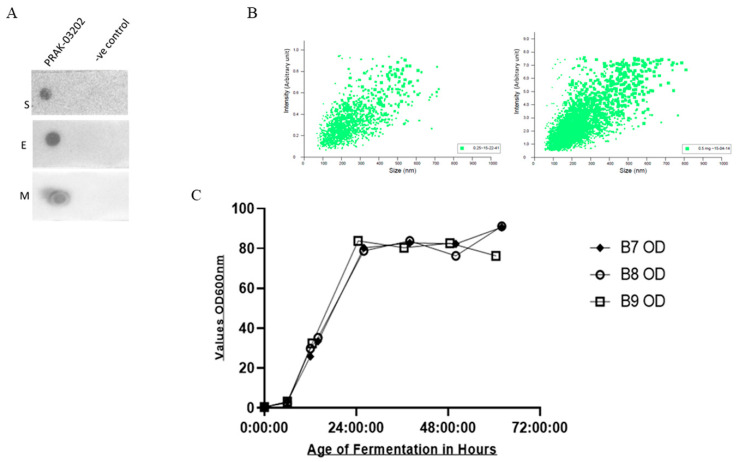
(**A**) Dot blot analysis for co-expression of spike, envelope, and membrane proteins in the *S. cerevisiae*-based D-Crypt™ platform using antigen-specific antibodies S: spike protein, E: envelope protein, M: membrane protein. (**B**) Nanoparticle tracking analysis with 0.25 (left panel) and 0.5 mg (right panel) of PRAK-03202. (**C**) Optimum time to obtain maximum production of PRAK-03202 in the fermenter. The fermenter was harvested at 24, 48, and 72 h to determine the maximum OD of the culture.

**Figure 2 vaccines-11-00743-f002:**
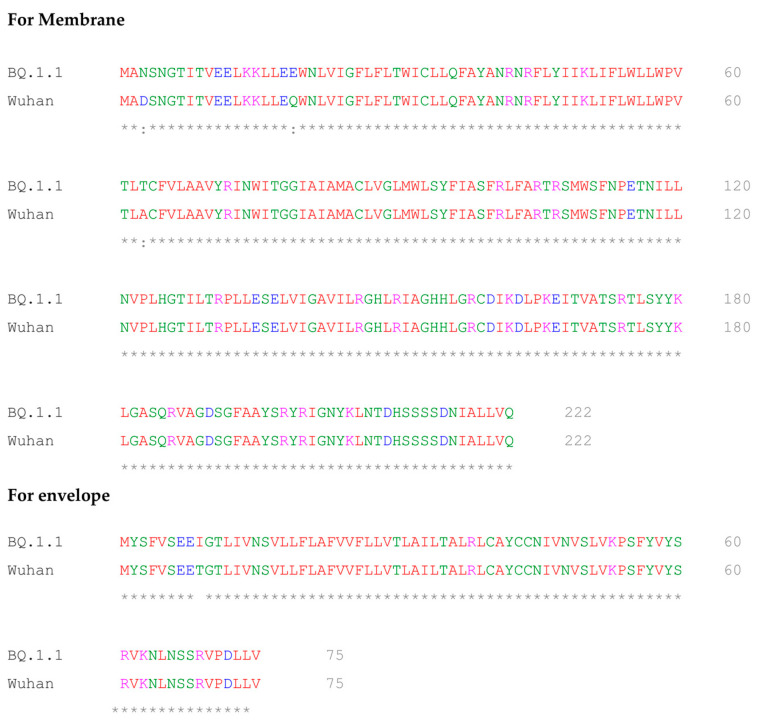
Multiple sequence alignment of BQ.1.1 and Wuhan variant of SARS-CoV-2. Fully conserved residues, insertion/deletion of amino acids, and non-conserved amino acids are indicated by *, ---, and a gap, respectively.

**Figure 3 vaccines-11-00743-f003:**
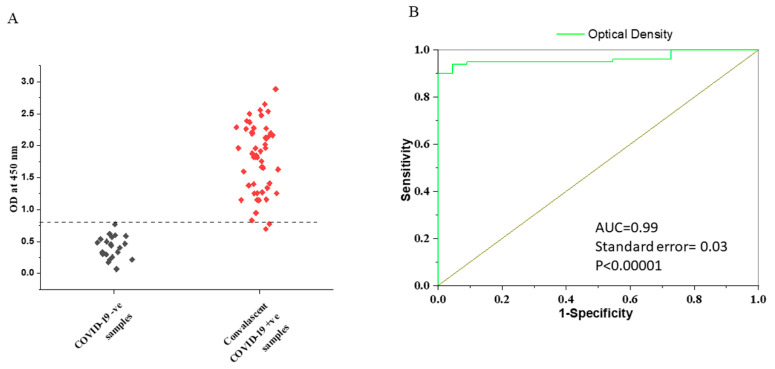
(**A**) IgG response against PRAK-03202 in COVID-19-negative and -positive convalescent samples. ---- indicates cut-off values. (**B**) ROC curve for the VLP-ELISA using optical density values.

**Figure 4 vaccines-11-00743-f004:**
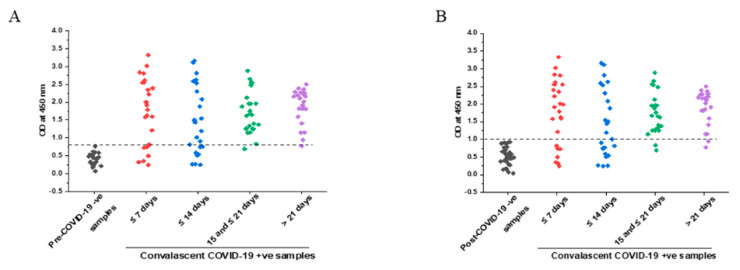
IgG response against PRAK-03202 in (**A**,**B**) COVID-19-positive convalescent samples at the indicated periods of symptoms onset and negative samples from (**A**) pre-COVID-19 and (**B**) post-COVID-19 era. ---- indicates the cut-off values.

**Table 1 vaccines-11-00743-t001:** Concentration of S, E, and M protein in PRAK-03202.

Antibody	Amount of Protein (ng)	Mol. Wt. (kDa)	Number of Molecules
RBD	105 ± 16	420 (trimer)	1.4 × 10^11^ ± 0.32
E-specific	10.25 ± 2.12	54 (pentamer)	1.1 × 10^11^ ± 0.3
M-specific	62.5 ± 1	25	1.51 × 10^12^ ± 0.05

**Table 2 vaccines-11-00743-t002:** Specifications of PRAK-03202.

SN	Test	Method	Specification
1	Appearance	Visual	Whitish suspension to clear liquid, free from foreign particles, and in which the mineral carrier tends to settle down on keeping
2	pH	pH meter	6.5–7.5
3	Total protein content	Bradford	Not less than (NLT) 10 µg/dose
4	Identity test	ELISA	Positive for S, M and E antibodies
5	Purity	Size exclusion chromatography–HPLC	>98% Pure
6	Endotoxin	Kinetic chromogenic method	Not more than (NMT) 10 EU/µg
7	Sterility	Membrane Filtration	No evidence of microbial growth
8	Abnormal toxicity test	Test method as per IP 2018	None of the animals die or show signs of ill health in 7 days following injection
9	Extractable volume	By pipetting	Not less than nominal volume of 0.5 mL
10	Sucrose estimation	Enzymatic method	NMT 15% w/v
11	Aluminum content (Al^3+^)	ICP-MS	NMT 800 µg/dose

**Table 3 vaccines-11-00743-t003:** Sensitivity, accuracy, positive, and negative predictive value of the VLP-ELISA assay to detect IgG antibodies in 23 rRT-PCR negative (pre-COVID era), 32 rRT-PCR negative (post-COVID-era), and 100 COVID-19-positive samples.

	**Days after COVID-19 Symptoms Onset (Pre-COVID Era)**			
**PRAK-03202**	**0–7 Days (95% CI)**	**8–14 Days (95% CI)**	**15–21 Days (95% CI)**	**≥22 Days (95% CI)**
True-positive patients (*n* = 25/each group)	18	15	24	24
True-negative (*n* = 23)		22		
Sensitivity	72% (51–88%)	60% (38–78%)	96% (80–99%)	96% (80–99%)
Specificity	96% (78–99%)	96% (78–99%)	96% (78–99%)	96% (78–99%)
Accuracy	83% (70–92%)	77% (63–88%)	97% (86–99%)	97% (86–99%)
Positive predictive value	94% (72–99%)	94% (68–99%)	96% (78–99%)	96% (78–99%)
Negative predictive value	76% (62–86%)	69% (57–78%)	97% (76–99%)	97% (76–99%)
	**Days after COVID-19 Symptoms Onset (Post-COVID Era)**			
**PRAK-03202**	**0–7 Days (95% CI)**	**8–14 Days (95% CI)**	**15–21 Days (95% CI)**	**≥22 Days (95% CI)**
True-negative (*n* = 32)		32		
Sensitivity	72% (51–88%)	60% (38–78%)	92% (89–100%)	92% (89–100%)
Specificity	100% (89–100%)	100% (89–100%)	100% (74–99%)	100% (74–99%)
Accuracy	88% (76–94%)	82% (70–91%)	96% (88–98%)	96% (88–98%)
Positive predictive value	100%	100%	100%	100%
Negative predictive value	82% (76–94%)	76% (66–84%)	94% (80–98%)	94% (80–98%)

**Table 4 vaccines-11-00743-t004:** End-point dilution titers of IgG antibodies to PRAK-03202 in a subset of early and later phase samples.

Serial Number	Sample ID	Time Since Symptom Onset	IgG Reciprocal Titer PRAK-03202
1.	A1002	≤14 Days	50,000
2.	A1003	≤14 Days	10,000
3.	A1004	≤14 Days	10,000
4.	A1005	≤14 Days	10,000
5.	A1007	≤14 Days	50,000
6.	A1012	≤14 Days	10,000
7.	A1021	≤14 Days	10,000
8.	A1030	≤14 Days	10,000
9.	A1039	≤14 Days	10,000
10.	A1041	≤14 Days	10,000
11.	A1045	≤14 Days	10,000
12.	A1051	>14 Days	10,000
13.	A1052	>14 Days	10,000
14.	A1075	>14 Days	50,000
15.	A1076	>14 Days	50,000
16.	A1078	>14 Days	50,000
17.	A1079	>14 Days	10,000
18.	A1091	>14 Days	50,000
19.	A1092	>14 Days	50,000
20.	A1093	>14 Days	50,000
21.	A1094	>14 Days	10,000

## Data Availability

The data presented in this study are available on request from the corresponding author. The data are not publicly available due to personal information about the patients.

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
