# Peer review of "VLP-ELISA for the Detection of IgG Antibodies against Spike, Envelope, and Membrane Antigens of SARS-CoV-2 in Indian Population"

_vaccines, 2023, doi:10.3390/vaccines11040743_

Round 1

Reviewer 1 Report

Manuscript ID: vaccines- 2254760

 Title

 VLP-ELISA for the detection of IgG antibodies against spike, 2 envelope, and membrane antigens of SARS-CoV-2 in Indian 3 population

 Authors: Dilip Kumar1, Sourav Singha Roy1, Ruchir Rastogi1, Kajal Arora1, Avinash Undale1, Reeshu Gupta1*, Nupur Mehro-5 tra Arora1 and Prabuddha K Kundu1*

This study investigates what the Authors consider an alternative in large surveys to Real-time reverse transcription–polymerase chain reaction (rRT-PCR), the gold standard for diagnosing severe acute respiratory syndrome coronavirus 2 (SARS-CoV-2) infection. They develop an ELISA serological method based on detection of IgG using “PRAK-15 03202” (PRAK), a virus-like particle (VLP), containing the spike(S), envelope (E), and membrane(M) proteins of SARS-CoV-2 in a highly characterized S. cerevisiae-based D-Crypt platform.

They evaluate sensitivity of their VLP-ELISA in 100 patients positive for SARS-CoV-2 by rRT-PCR in naso-/oro-pharyngeal swabs. They also use non-SARS-CoV-2 serum a small number of samples collected before and after the COVID-19 pandemic to evaluate specificity. Moreover, they confirmed that sensitivity, specificity, and accuracy were 96% in 30 samples collected >14 days after symptom onset. The Authors conclude they successfully developed an in-house VLP ELISA to detect IgG antibodies against three antigens of SARS-CoV-2 as a simple, affordable, and feasible alternative test for sero-epidemiological surveys.

 My main objections are:

1)      There is some confusion in the introduction/abstract/discussion between positivity to infection (asymptomatic or not) and positivity to antibodies (which includes patients that have recovered from the disease (anti S, M, E, N etc IgG), vaccinated patients (anti S IgG) and, based on what the Authors state, patients at 14 days from symptom onset. This makes it unclear as to what the test developed in the present work is going to be used for, and I think the Authors should state this better. The test cannot be intended for COVID diagnosis because 1) IgG is present in sera of people who have recovered from the disease 2) IgG anti PRAK can only be found reliably after day 14 from disease onset (roughly 20 days after infection), making antigenic test more reliable and efficient as an alternative to RTPCR. It cannot discriminate between recovered patients and patients vaccinated. In line 495, for example, they claim the test might be used to help safe mobility and do not explain how this test would help more that looking for anti S IgG, as currently done. Lines 488-501 of the discussion should be made clear, as at the moment the test described does not seem to bring any advantage compared to the ones on market.

2)      A help in defining the issue above would come from comparing this test to a simple anti SIgG search. There is no comparison of results of ELISA on PRAK to those obtained with the single antigens mentioned in line 108. What is the advantage of this ELISA over the ones already developed and on market, detecting, for example, antiN IgG?

3)      Why is figure 4C in Figure 4? This is more related to results in Fig.1 and has nothing to do with Fig. 4 A and B. Also Table 4 should be in the part describing the preparation of PRAK, as done in the first paragraph od results.

 Minor comments

1) A brief explanation of what D-Crypt platform is would be helpful.

2) It would also be nice to see an SDS PAGE analysis of their D-Crypt-S-E-M product, in addition to the NTA analysis, which in my opinion does not add significantly to their data.

3) Source of Purified proteins (S, M, and E) used to titrate PRAK (line 108)?

4) Sequence comparison of BQ.1.1 and Wuhan variant of SARS-CoV-2. It is not clear why the Authors are performing this analysis. Please explain better.

5) Line 380: is this an OD value? please state. This value is quite high as a cutoff: does this mean there is reactivity in the preCOVID sera (possibly due to immune responses against other coronaviruses)? There is not much difference between IgG in <7 day samples compared to >14 days, possibly due to anamnestic antibody responses.

6) Line 394:  delete “for”

7) Why is figure 4C in Figure 4? This is more related to results in Fig.1 and has nothing to do with Fig. 4 A and B. Also Table 4 should be in the part describing the preparation of PRAK

Line 443: “The involvement of membrane proteins in recognizing IgG, IgM, and IgA in SARS-CoV-2-infected patients” should be “The ability of IgG, IgM, and IgA to bind membrane proteins in SARS-CoV-2-infected patients”.

Line 445. Chose not Choose

Line 150: better specify horseradish peroxidase (HRP)-conjugated anti-human IgG secondary antibodies

Line 152: insoluble, should be soluble.

My conclusions:

 The study is well conducted and clear results were shown. The text is well written. However, the authors should address the issues outlined above.

Author Response

Dear Sir/Ma'am

Thank you.

Reviewer 2 Report

The publication presents the development and evaluation of Covid VLPs representing the M, E and S protein of SARS-CoV-2 for the use in serological assays for detection oof specific IgG.

In principle the use of multiple antigens could increase the sensitivity.

The manuscript clearly describes the work and procedures used for this study. However, the number of sera used, and kind of investigation limit the study as pre-evaluation of an in-house assay.

Some specific comments:

The characterization of the antigen should be clearly separated from the assay evaluation.

Fig 1A., Table 1, Figure 2, Figure 4c belongs to the antigen production and characterization.

Fig. 1A: A dot blot is not the best method to analyze the quality and purify of an antigen. A SDS PAGE with a specific immunoblot staining would be much more specific.

If the specific antisera against S, M, and E protein would react with impurities of the fermenter grown material this could be not recognized in a dot blot.

Table 3: The IgG titers are reciprocal titers? Which should be clearly described.

Since the purity of the PRAK-03202 preparations are unclear a proper control of a yeast debris would be recommended for the analysis of the patient sera.

Table 2A: The validity of data would benefit from real numbers x of y instead of pure % values. 23 and 100 are a moderate number of samples for an assay evaluation if divided in four categories.

General comments:

The evaluation would significantly benefit from comparison the new developed assay directly with one or two other commercial assays detecting anti-COVID antibodies.

Since the antibody response in the acute phase starts with IgM which is not targeted in this assay. Therefore, the assay missed an important target for an early detection of a COVID infection.

Also, the authors have completely neglected unspecific reaction by other Corona viruses and other pre-existing infections or diseases causing false positive results in ELISAs. Malaria, autoimmune, and rheumatic disease could generate very unspecific reactive antibodies.

All these issues should be mentioned and discussed in more detail as they limit the use of the current test significantly. All these evaluations must be performed before this assay should be used for any studies analyzing patient’s immune response.

Author Response

Dear Sir/Ma'am

Thank you.

Round 2

Reviewer 1 Report

The Authors have made considerable efforts to improve their manuscript.

Reviewer 2 Report

The manuscript has signifantly improved by the authors. However such statement:   "However, the sensitivity of the current VLP-ELISA is greater in the first and second weeks of symptom onset than that of other existing assays." has to be demonstrated or deleted.

I suggest the acceptance of the manuscript after this final change.

Author Response

Dear Sir/Ma'am

Thank you very much for your valuable inputs.

Comment:

The manuscript has significantly improved by the authors. However, such statement:   "However, the sensitivity of the current VLP-ELISA is greater in the first and second weeks of symptom onset than that of other existing assays." has to be demonstrated or deleted.

Response: As suggested, line has been deleted (Line: 521-523).

Thank you.

Regards.